# Effect of Fluorographene Addition on Mechanical and Adhesive Properties of a New Core Build-Up Composite

**DOI:** 10.3390/polym14235301

**Published:** 2022-12-04

**Authors:** Paolo Baldissara, Davide Silvestri, Giovanni Maria Pieri, Claudia Mazzitelli, Antonio Arena, Tatjana Maravic, Carlo Monaco

**Affiliations:** 1Department of Biomedical and Neuromotor Sciences, Alma Mater Studiorum, University of Bologna, 40125 Bologna, Italy; 2Department of Surgery, Medical, Dentistry and Morphological Sciences with Transplant Interest, Oncology and Regenerative Medicine (CHIMOMO), University of Modena and Reggio Emilia, 41124 Modena, Italy

**Keywords:** dental composite, dental resin cement, core build-up material, 10-MDP, graphene, fluorographene, mSBS, flexural strength, zirconia adhesion, dentin adhesion

## Abstract

This study aims to develop a restorative material having such mechanical and adhesive properties that it can be used both as a reconstruction material and as a luting cement. The experimental core build-up composite (CBC) was derived from a self-adhesive cement by the modification of its chemical formula, requiring the use of dedicated dentin and ceramic primers. The adhesive properties to zirconia and dentin were analyzed with a micro-Shear Bond Strength test (mSBS). The mechanical properties were analyzed by a flexural strength test. The results were compared with those obtained for other commercially available cements and core build-up materials, both before and after addition of 2 wt.% fluorographene. The CBC obtained average values in the mSBS of 49.7 ± 4.74 MPa for zirconia and 32.2 ± 4.9 MPa for dentin, as well as values of 110.9 ± 9.3 MPa for flexural strength and 6170.8 ± 703.2 MPa for Young’s modulus. The addition of fluorographene, while increasing the Young’s modulus of the core build-up composite by 10%, did not improve the adhesive capabilities of the primers and cement on either zirconia or dentin. The CBC showed adhesive and mechanical properties adequate both for a restoration material and a luting cement. The addition of 2 wt.% fluorographene was shown to interfere with the polymerization reaction of the material, suggesting the need for further studies.

## 1. Introduction

The possibility of having a single dental restorative material used both for reconstructive and luting purposes presents many clinical advantages, especially for the reconstruction of endodontically treated teeth with or without fiber posts [1,2]. In this clinical scenario, this multipurpose material could be employed for the cementation of an endodontic post, for the reconstruction of a prosthetic abutment, and for the cementation of a prosthetic crown to form a monoblock unit [3], simplifying clinical procedure and reducing operator-dependent errors [2]. In order to fulfill these requirements, the material should possess specific mechanical and rheological characteristics.

The literature reports that the addition of nanometric fillers to dental adhesive polymers can improve bonding properties to dentin due to higher mechanical properties of the adhesive layer and stronger interface interaction with the substrate [4,5,6,7]. In the same way, nanometric fillers added to a resin matrix were shown to increase the mechanical properties of composites [8].

In recent years, graphene and its derivatives have been extensively studied to improve the mechanical properties of nanocomposites in several fields of application [9,10]. Since the discovery of graphene in 2004 [11], many studies have been conducted on its integration into polymer matrices [9,10,12]. Its mechanical properties reach 1 TPa for Young’s modulus, and it has an intrinsic strength of 130 GPa, values that assert graphene as the strongest material ever tested [13]. The presence of particles with Young’s modulus values higher than those of resin matrices hinders the formation and propagation of cracks thanks to the crack branching, crack bridging, pull-out, and crack deflection phenomena [14,15].

In the literature, it has been described how the addition of graphene derivates can increase mechanical properties in Portland-type cement [16], bioactive calcium silicate cements [17], epoxy resins [18], nanocomposites based on bisphenol A-glycidyl methacrylate/tetra(ethylene glycol) diacrylate (Bis-GMA/TEGDA) [19,20], ceramics [15], and glass-ionomer cements [21]. The addition of graphene derivatives to dental adhesives has been shown to improve adhesive properties [22,23,24,25], antibacterial properties [26,27], and adhesion durability [28].

However, due to the dark color of graphene [29] and inherent aesthetic drawbacks, its application in modern restorative dentistry is quite difficult and is restricted to thin-layer materials, such as dentin adhesives [7]. Instead, fluorographene (FG), a fluorine-functionalized graphene, may have great potential for application in the dental field because of its bright white color when the fluorine is close to the full saturation level [29].

Fluorographene is a derivate of graphene sharing the same morphological and structural pattern but with lower mechanical properties due to presence of fluorine, which creates imperfections in the lamellar structure [29,30]. The high mechanical and antibacterial properties in conjunction with the light color (provided that the concentration of fluorine is up to 50 wt.%) make fluorographene a promising nanofiller for dental materials, as has been reported by several authors [21,31,32]. Different concentrations of fluorine within fluorographene also affect the hydrophobicity of the material itself, leading to increased hydrophobicity at higher fluorine concentrations [29].

In light of the aforementioned considerations, the purposes of this study are to evaluate the effect of a 2 wt.% fluorographene addition on the adhesive and mechanical properties of an experimental core build-up material and primer system with dentin and zirconia substrates.

In particular, the null hypotheses tested (α = 0.05) are as follows:
The adhesive and mechanical properties of the experimental CBC system are not significantly different from those of other existing luting cements and core build-up materials;The addition of 2 wt.% fluorographene to both dentin and ceramic primers has no significant effect on adhesive properties to zirconia and dentin;The addition of 2 wt.% fluorographene has no significant effect on the flexural properties of the CBC;The addition of 2 wt.% fluorographene to both dentin and ceramic primers has no significant effect on the mSBS of the CBC on zirconia and dentin when compared to the controls.

## 2. Materials and Methods

Industrial fluorographene was purchased from ACS Material (Pasadena, CA, USA) and observed with a scanning electronic microscope (SEM-FEG Nova NanoSEM 450, FEI, Eindhoven, The Netherlands) for morphological characterization (Figure 1). 

The manufacturer declared a fluorine content of at least 53 wt.%, a flake diameter of 4–10 µm, and a flake thickness of 5–10 nm. Two different composite and primer material systems were prepared using the basic chemistry derived from a self-adhesive luting cement (OverCEM SA, Overfibers Srl, Imola, Italy): (1) a core build-up composite (CBC) prototype with dispersed, short glass fibers (200 µm) and (2) a CBC prototype derived from the former by adding 2 wt.% fluorographene (FG) to the resinous fraction. The prototypes had their own dentin (DP) and ceramic (CP) primers that were also made without 2 wt.% FG addition (Table 1).

### 2.1. Graphene-Reinforced CBC Preparation

Fluorographene was added to the CBC prototypes in two different forms: (1) micronized and (2) exfoliated.

#### 2.1.1. Micronized FG/CBC (CBC 2% FGm)

Micronization of the FG was accomplished in a ball-milling jar with 0.6 mm zirconia spheres and isopropyl alcohol as a milling medium. After 3 h at 320 rpm, the FG was separated from the spheres, and the alcohol was evaporated in a stove at 37 °C for 24 h. This micronized FG was added to the CBC base paste at a concentration of 2 wt.% calculated based on the resinous fraction. Then, it was processed in an orbital mixer (Speedmixer 800, Hauschild, Germany) at 1500 rpm for 1 min.

#### 2.1.2. Exfoliated FG/CBS (CBC 2% FGe) and Dentin and Ceramic Primers (DP 2% FG; CP 2% FG, Respectively)

The exfoliation procedure exploited the method described by Zhu et al. [33]. Briefly, 1.50 g of industrial FG was dispersed in 240 mL of chloroform (250 mg of FG dispersed per 40 mL of chloroform). The solution was then ultrasonicated (Hielscher UP200St, Hielscher, Germany) in an ice bath to maintain the temperature at 2–4 °C for about 5 h (frequency = 26 kHz, power output = 100–130 W, continuous 100% duty cycle at 150 µm oscillation amplitude using a titanium sonotrode of 14 mm in diameter. The solution was then placed in 40 mL jars and centrifuged at 3840 rpm in a centrifuge 190 mm in diameter to accelerate the sedimentation of the nonexfoliated, heavier particles, which were then separated and prepared for the next exfoliation procedure. After evaporation of the chloroform in air at 37 °C, residual powder of exfoliated FG was harvested and dispersed in pure acetone. The evaporation of liquid acetone was carried on until the desired concentration of exfoliated FG was obtained. The addition of exfoliated FG to the catalyst paste of the CBC was achieved by dispersion of 0.06 g of FG in 1.2 mL of acetone per 3 g of resin, and the acetone was left to evaporate for 24 h at 37 °C. The same FG-dispersed acetone suspension was used to make the FG-containing dentin (DP) and ceramic (CP) primers without any evaporation since the primers already required a certain acetone content.

### 2.2. Micro Shear Bond Strength (mSBS) Tests of CBC Prototype and Composite Luting Cements 

For mSBS on ceramic substrate, zirconia blocks (Katana STML A2, Kuraray Noritake, Tokyo, Japan) were used. At the pre-sintering stage, the zirconia was severed with a microtome (Micromet Evolution, Remet, Casalecchio di Reno, Italy) to obtain 15 × 19 mm^2^ plates of ~3 mm thickness. These plates were polished with SiC abrasive paper at 100, 240, and 400 grit in wet conditions and then were washed, dried, and sintered with a conventional sintering cycle (Z1800 Furnace, Forno Mab s.r.l., Ponte Sesto di Rozzano, Italy). Sintered plates were embedded in acrylic resin (Technovit 4071, Kulzer, Germany), polished with 400-grit abrasive paper in wet conditions, and subjected to air abrasion treatment with 50 μm of aluminum oxide (Cobra, Renfert, Germany) using a bar pressure of 3.5, a 90° angle, and a 10 mm distance. Surface analysis was carried out on each sample by means of a profilometer (Perthometer M4P, Mahr Perthen, Germany) to assess the final surface roughness (Ra). Then, the specimens were ultrasonicated in distilled water for 2 min to allow debris detachment and dried in air. 

Dentin specimens were obtained from freshly extracted, sound human third molars. The teeth were embedded in resin cylinders (Technovit 4071) and cut 4 mm below the cuspal-tip plane with a 0.35 mm thick diamond blade to expose the dentin surface. The specimens were polished with SiC sandpaper at 100, 240, and 400 grit in wet conditions, cleaned with a solution of 0.2% chlorexidine and 0.2% cetrimide (Clotramid, Molteni Farmaceutici SpA, Scandicci, Italy), and then rinsed with water spray. The DP and CP were exclusively applied when using the CBC prototypes and the Panavia V5 (Kuraray Noritake, Tokyo, Japan) luting cement since the other cements did not require any primer (self-adhesive materials). A complete list of the materials used in the study is showed in Table 2.

Then, several composite cylinders of 1.5 mm in diameter were made directly on both zirconia and dentin specimens in a hexagonal array using a dedicated, custom-made silicone rubber mold and a plexiglass pressure plate loaded with a force of 5 N. Light-curing irradiation for 60 s (Valo Cordless, Ultradent Products, Inc., South Jordan, UT, USA) was delayed for 1 min and 30 s to allow an undisturbed chemical reaction of the composite cylinders with the substrate surface.

Samples were placed in distilled water and stored at 37 °C for 24 h. Tests of mSBS were carried out with a universal testing machine (4301, Instron, Canton, MA, USA) at a cross-head speed of 0.1 mm/min (Figure 2).

### 2.3. Three-Point Flexural Test of Core Build-Up Composites 

Samples were made using a 25 ± 2 mm × 2.0 ± 0.1 mm × 2.0 ± 0.1 mm stainless-steel mold in accordance with the ISO 4049:2019 standard. The material was injected into the mold, starting from the edges and gradually filling the cavity to slight excess, avoiding the inclusion of air bubbles and voids. Dedicated mixing tips were used for each material. Table 3 shows the materials used for the mechanical tests.

Through compression carried out with an insulated glass plate, all excess cement was evacuated. The samples were polymerized with a 1000 mW/cm^2^ curing lamp (Valo Cordless, Ultradent, South Jordan, UT, USA), starting from the center and moving towards the ends, with sequences of 10 s each for both sides of the sample for a total of 50 s. The whole mold was placed in water at 37 °C for 15 min; then, the solid composite beam was removed, finished with 400-grit sandpaper, and kept in distilled water at 37 °C for 24 h until mechanical testing. 

Five specimens for each type of material were tested using a universal testing machine (2530-1KN, Instron, Canton, MA, USA) with a descending speed of 0.75 mm/min. Then, the means and standard deviations of both flexural resistance and flexural Young’s modulus were calculated.

### 2.4. Optical and SEM Analysis

Morphological analysis of the composite specimens and substrate materials after testing, as well as FG dispersion observations, were carried out with an optical stereomicroscope (Stemi 305, Zeiss, Aalen, Germany) at a maximum magnification of 40× using its dedicated imaging software (Zen 2). Solid specimens were directly observed after cleaning with a gentle stream of dry air. Liquid specimens were observed as drops placed on thin microscope observation glass. 

Pure industrial FG was observed with a scanning electron microscope (SEM-FEG Nova NanoSEM 450, FEI, Eindhoven, The Netherlands) to observe the original morphology of the starting material. Dentin specimens were fixated in a 2.5% glutaraldehyde solution and then dehydrated in ascending ethanol concentrations (50, 70, 80, 90, 95, and 100%) and hexamethyldisilazane. Both dentin and zirconia specimens were sputter-coated with gold and observed under SEM (JEOL, Tokyo, Japan).

### 2.5. Statistical Analysis

Statistical analysis was performed using GraphPad Prism 9 (GraphPad Software, San Diego, CA, USA) statistical software. D’Agostino–Pearson normality tests (α = 0.05) were carried out to evaluate the distribution of values.
Adhesion (mSBS) analysis: Due to the non-normal distribution of the data, the comparative analysis among the CBC prototypes and the luting cements was performed using a nonparametric ANOVA (Kruskal–Wallis test) and Dunn’s multiple comparisons test (α = 0.05). The effect of 2 wt.% FG added to DP and CP was analyzed using an unpaired *t*-test with Welch’s correction;Mechanical properties analysis: Because of the non-normal distribution recorded in every group, Kruskal–Wallis and Dunn’s multiple comparisons tests were used (α = 0.05) to compare the CBC prototype to the already-existing CBC materials, as well as to the CBC after the addition of 2% FG.

## 3. Results

### 3.1. Micro Shear Bond Strength (mSBS) Tests of CBC Prototype and Composite Luting Cements 

The results of the mSBS tests are given in Table 4. The Kruskal–Wallis test showed a difference that was statistically significant among the medians of the CBC and other cements (*p* < 0.0001), both on zirconia and dentin.

#### 3.1.1. Zirconia

No statistical difference between the CBC prototype and OverCEM SA was found (*p* = 0.7079). The CBC reached statistically higher adhesion values than the other tested cements (*p* < 0.0001, Dunn’s multiple comparisons test).

An analysis of the results reached by materials with addition of FG showed that, on zirconia, CBC + CP 2% FGe possessed significantly lower adhesion strength than the CBC alone (*p* < 0.0001), while CBC 2% FGe had no statistical differences from the CBC alone (*p* = 0.1520).

#### 3.1.2. Dentin

On dentin, Dunn’s multiple comparisons test highlighted no statistical difference between the CBC and Panavia V5 (*p* = 0.9999). The CBC reached statistically higher adhesion values than the other tested cements (*p* < 0.05). 

On dentin, CBC + DP 2% FGe possessed significantly lower adhesion capacity than the CBC alone (*p* < 0.0001).

A graphical analysis is shown in Figure 3.

### 3.2. Flexural Strength Test

The results of the flexural strength test are given in Table 4. 

The Kruskal–Wallis test for flexural strength reported no statistically significant differences among the medians of the CBC and the other tested core build-up materials (*p* = 0.2056). 

Statistically significant differences in Young’s modulus were found among the tested materials with the Kruskal–Wallis test (*p* = 0.002). Dunn’s multiple comparisons test highlighted a significant difference between the Young’s modulus values of the CBC and Bisfil 2B (*p* = 0.0091).

### 3.3. Effect of Fluorographene

The addition of 2% micronized FG to the CBC increased the Young’s modulus values of the material by 10%, but the variations were not statistically significant in either flexural strength (*p* = 0.5574) or Young’s modulus (*p* = 0.2084).

After the addition of 2% exfoliated FG to the CBC, a significant worsening of flexural strength was observed (*p* = 0.0048). No statistically significant differences were observed in the Young’s modulus values (*p* = 0.1783) (Figure 4).

### 3.4. Morphological and Fractographic Analysis

The optical microscope analysis of the experimental CBC showed a correct dispersion of the fillers inside the matrix in both the control sample and FG-containing CBC samples (2% FGm and 2% FGe CBCs; Figure 5).

The presence of air bubbles was visible in the cement matrices prior the degasification process. A morphological analysis with SEM showed a prevalence of adhesive fractures on zirconia and the presence of adhesive and cohesive fractures on dentin (Figure 6).

## 4. Discussion

The purpose of this study was to produce a CBC with adhesive properties that could be used both as a restorative material and a luting cement. The CBC prototype was made starting from a commercially available self-adhesive resin cement (OverCEM SA) by reducing the content of 10-methacryloyloxydecyl dihydrogen phosphate (10-MDP, an adhesive monomer) and hydroxyethyl methacrylate (HEMA, a hydrophilic monomer), which are used to facilitate the infiltration of wet dentin by acrylic, polymerizable molecules [34].

The above-mentioned monomers were substituted by hydrophobic, cross-linking diacrylates, such as Bis-GMA, TEGDMA, and UDMA, with the aim of improving the reticulation degree, the mechanical properties, and the hydrolytic stability of the resulting composite in accordance with the structural requirements of a restorative CBC material. The formulation of the fillers was also modified: glass fibers with an aspect ratio of ~20:1 (200 μm × 10–12 μm) were introduced to increase the fracture resistance properties through a randomly distributed crack-bridging effect [35].

Since the self-adhesive properties of the CBC prototype were reduced by removing the adhesive monomer content, it was necessary to introduce the 10-MDP monomer in two separate primers to maintain high adhesion values between the new composite and the dentin and ceramic substrates [34,36].

Indeed, when added to resin adhesives and self-adhesive luting cements, the 10-MDP monomer induces a relevant increase in the strength and stability of dentin and zirconia ceramic bonding [2,37]. On dentin, paired 10-MDP molecules joined by stable MDP-Ca salt formation generate arrays of self-assembled, nano-layered structures that are responsible for long-term adhesion. This bonding mechanism occurs without depleting the hydroxyapatite crystals from the collagen fibrils [37,38], hindering the collagen degradation phenomena observed with etch and rinse dentin adhesives [39,40] that, conversely, require collagen exposure to generate bonding.

On zirconia, an increasingly used ceramic for restorative and prosthetic dentistry, the 10-MDP bonding mechanism occurs through phosphoric group interaction with the zirconium oxide substrate with the formation of both hydrogen and ionic bonding and, possibly, lateral bonding between neighboring 10-MDP phosphate groups [41].

Searching for further improvements of the mechanical properties of the CBC prototype, which are highly desirable in a structural dental composite, a derivative of graphene, fluorographene (FG), was chosen as a nanofiller reinforcement and added to the CBC in both micronized and exfoliated forms.

Exfoliated FG was also added to the CBC-associated primers with the aim of improving their adhesion strength to both dentin and zirconia ceramics. Zirconia, in particular, is one of the most difficult restorative dental ceramics to be adhered, even with the most advanced dental luting cements. The rationale for the use of FG in the ceramic and dentin primers was to exploit the mechanisms of crack branching, crack bridging, pull-out, and crack deflection that nanofillers such as graphene and FG can develop to increase the strength of the weak, thin resin polymer layer that forms when primers are applied to a substrate. Evidence of the improvements generated by graphene-derived compounds added as a filler in several dental materials (dental adhesive primers, luting cements, and restorative composites) has been reported in the recent literature [18,21,22,23,24,25,26,31].

FG was also chosen because of its light color, which is at a maximum at the highest fluorine saturation. Color is an important property in restorative material since it can influence the aesthetics of the final restoration. The FG powder employed in this study showed grains with different fluorination degrees, resulting in an overall gray color, with white and almost black fractions mixed together. However, since this research was mostly focused on the theoretical contribution of FG to the structural resistance of the CBC, a nonoptimal colored FG mixture was accepted to make the CBC prototypes. Further studies should be focused on the correct color balance by selecting the most fluorine-saturated, white-colored form of FG.

The dispersion procedures of industrial FG inside the CBC and primer prototypes were characterized by a fine-tuning phase due to the strong tendency of the filler to aggregate into clusters. In pre-experimental tests, mechanical mixing showed an unacceptable dispersion, and the resulting composite was, therefore, discarded. An industrial FG dispersion procedure was then carried out using an orbital mixer (Speedmixer 800) after a FG micronization process in a ball-milling grinding machine, which was more satisfactory in terms of material homogeneity (CBC 2% FGm). To further improve the exfoliation and dispersion of FG, the procedure described by Zhu et al. [33] was adapted and applied, which allowed the addition of exfoliated FG dispersed in acetone to the ceramic and dentin primers. 

The addition of exfoliated FG to the CBC base paste either by orbital mixing or using acetone as a dispersion medium led to a fast polymerization reaction of the material, suggesting a supporting role of FG for the CBC catalysts, a chemical interaction with other substances contained in the composite, or even a direct catalytic role [42,43]. As reported by Yam et al. [44] for graphene, due to the one-atom thickness and zero band-gap with low density of states around the Fermi level, graphene could have great potential roles in various catalysis applications. Since the FG used in this study was a mixture of differently fluorine-saturated forms, a certain quantity of graphene could be present and be responsible of the additional catalytic effect here recorded. Probably, the mixture of saturated and unsaturated FG acted on the CBC initiators through the graphene fraction, in particular on the benzoyl peroxide (BPO), which provided the autopolymerizing reaction of the experimental composite.

Three different experimental groups were tested for their mSBS on zirconia and dentin to evaluate their adhesive properties: CBC + primers, CBC + 2% FG primers, and CBC 2% FGe + control primers (Table 4). However, the latter was not tested on dentin due to the early polymerization of the CBC material mentioned above, which reduced the working time necessary to make the mSBS cylinders. 

Then, three different experimental groups were tested with a flexural strength test to evaluate their mechanical properties: CBC, CBC 2% FGm, and CBC 2% FGe. CBC 2% FGm was only tested for mechanical properties to evaluate the effects of the different procedures of FG dispersion; it was not used in the mSBS comparison with luting cements since its color was unacceptably dark for a current luting material.

A comparison of the CBC properties with the ones of other commercial resin luting cements and core build-up materials was carried out to establish whether the adhesive and flexural properties were similar to those of already-existing materials. The adhesive strength of the CBC prototype was the highest among the different luting cements on both the zirconia and dentin substrates. With the exceptions of OverCEM SA on zirconia and Panavia V5 on dentin, all the other materials showed a statistically significant difference from the CBC (*p* < 0.0001). On the basis of the results obtained, the first null hypothesis could be only partially accepted since OverCEM SA on zirconia and Panavia V5 on dentin reached adhesion values close to that of the CBC. It is remarkable that, on zirconia, a self-adhesive cement (OverCEM SA) showed mSBS values even higher than that of Panavia V5, a material that belongs to the category of resin cements with separate adhesive systems, which are generally superior to self-adhesive cements regarding bonding to polycrystalline ceramics. Considering the outstanding in vitro adhesive strengths showed by the CBC prototype, this composite could be an excellent starting point for a definitive luting cement able to strongly bond zirconia to dental substrates. The mSBS of the CBC prototype was indeed at the highest level ever reported in the recent literature [45,46,47], but it must be remembered that no aging treatments were applied to the specimens tested; thus, a drop in mSBS values is likely to happen once the material undergoes long-term water storage or thermocycling regimens. Further studies are necessary to confirm the residual bond strength of the CBC in simulated clinical conditions.

As regards the flexural properties of the CBC prototype, no significant differences were found when compared to the other already-existing core build-up composites in terms of flexural strength (*p* = 0.206), although the CBC showed significantly lower values of Young’s modulus than Bisfil 2B (*p* = 0.009). The Young’s modulus of a resin cement is an important parameter since metal-free restorations need a stiff cement layer to reduce tensile strains at the intaglio surface, hindering the insurgence of radial cracks that lead to restoration failure [48]. 

Due to the exception of the significantly higher Young’s modulus of Bisfil 2B, the second null hypothesis was partially accepted, although in general, the mechanical properties of the CBC were proved to be not statistically different from those of commercially available core build-up materials. These results suggest that the CBC prototype could be used as a core reconstruction material, as far as flexural properties are concerned.

The addition of micronized FG to the CBC was shown to have no effect on either the flexural strength or the Young’s modulus, although an increase of 10% in Young’s modulus, still not statistically significant, was observed. The addition of 2% exfoliated FG showed a significant worsening of the flexural strength of the CBC (*p* = 0.0048); accordingly, the third null hypothesis was rejected. All these unexpected results suggest that the exfoliation procedures and inclusion techniques of FG in resin matrices should be perfected and deeply analyzed in order to obtain increases in composite mechanical properties. 

One concern about the addition of FG to the CBC prototype is the toxicity potential that could emerge from its use in restorative dental material. Although a toxicity assessment was not completed in this study, the literature has reported some contrasting data. Romero-Aburto et al. [49] reported that the incubation of FG with human breast cancer cell line MCF-7 did not show any cytotoxicity effects, even after 3 days, using a concentration of 576 μg/mL. Conversely, Teo et al., in two more recent studies [50,51] reported a dose-dependent cytotoxic effect of FG and other fluorinated nanocarbon materials. On human lung cancer cell line A549 after 24 h at concentrations of 12.5 μg/mL, the cell viability was between 100% and 95%, depending of the type of FG nanomaterial; with a higher concentration (400 μg/mL), the cell viability dropped to values ranging from ~85% to 5%. The data from these studies suggest that cytotoxic effect, although largely variable, is also dependent on the size and shape of fluorine-containing groups present in the nanomaterial, with higher fluorine content seeming to be related to higher cytotoxic effects.

The absolute percentage of FG used in the experimental CBC was 2 wt.% calculated based on the resin matrix fraction, and this quantity corresponded to ~0.7% of the whole CBC mass. Considering that the mean solubility recently reported for two resin luting cements [52] ranged from 3.20 to 5.41 µg/mm^3^, depending on the pH value of the artificial saliva medium, it seems unlikely that the percentages used in the CBC prototype could generate any serious adverse biological effect. Further studies are necessary to establish the real risk related to the use of FG in dentistry.

The addition of exfoliated FG to the CBC primers had significant detrimental effects on the adhesion; consequently, the fourth null hypothesis was rejected. FG was shown to significantly worsen the adhesion to zirconia and dentin when added to both the CP (*p* < 0.0001) and DP (*p* < 0.0001). The expected reinforcement of the thin primer layer and the interactions with the substrate operated by the FG inclusion did not properly occur. It could be hypothesized that the rearrangement of FG in clusters of excessive size caused weakening of the binding energy in the microstructure when the concentration became too high [20]. In addition, the phenomenon of early polymerization observed during the dispersion of exfoliated FG inside the base paste of the core build-up composite also suggest an incorrect crosslinking of the materials when they came in contact with exfoliated fluorographene. Recently, Lai et al. [53] reported that FG could directly initiate the highly efficient free-radical polymerization reaction of a styrene monomer with a high yield of free polystyrene of high molecular weight. Furthermore, the FG initiator seems to possess a long lifetime of chain radical centers and insensitivity to molecular oxygen; the latter is a very interesting property for dental composite material if an FG initiator role is also confirmed for acrylate monomers. 

## 5. Conclusions

–The experimental CBC tested in this study associated with ceramic primer and dentin primer based on 10-MDP showed adhesive and mechanical properties compatible for use both as a cement and as a core build-up material;–The addition of micronized fluorographene to the CBC at a percentage of 2 wt.% of the resin fraction increased the elastic modulus by about 10% but had no significant effects on flexural strength;–The addition of exfoliated fluorographene at a percentage of 2 wt.% to the 10-MDP dentin and ceramic primers reduced the adhesive strength of the CBC and ceramic primer system when applied to zirconia;–The addition of 2% exfoliated fluorographene to the core build-up composite reduced both the flexural strength and the Young’s modulus values;

The results suggest an interesting activity of fluorographene as a polymerization initiator or promoter; this finding, which to our knowledge has never been described for dental acrylates, strongly encourages further studies.

## Figures and Tables

**Figure 1 polymers-14-05301-f001:**
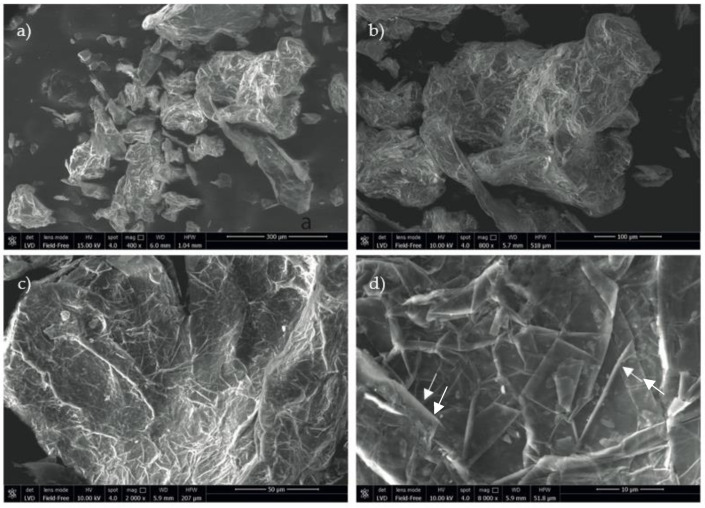
SEM images of industrial fluorographene at different magnifications: (**a**) 400×; (**b**) 800×; (**c**) 2000×; (**d**) 8000×. Typical fluorographene morphology showing a multilayered structure and characterizing grain morphology at higher magnifications (white arrows in (**d**)).

**Figure 2 polymers-14-05301-f002:**
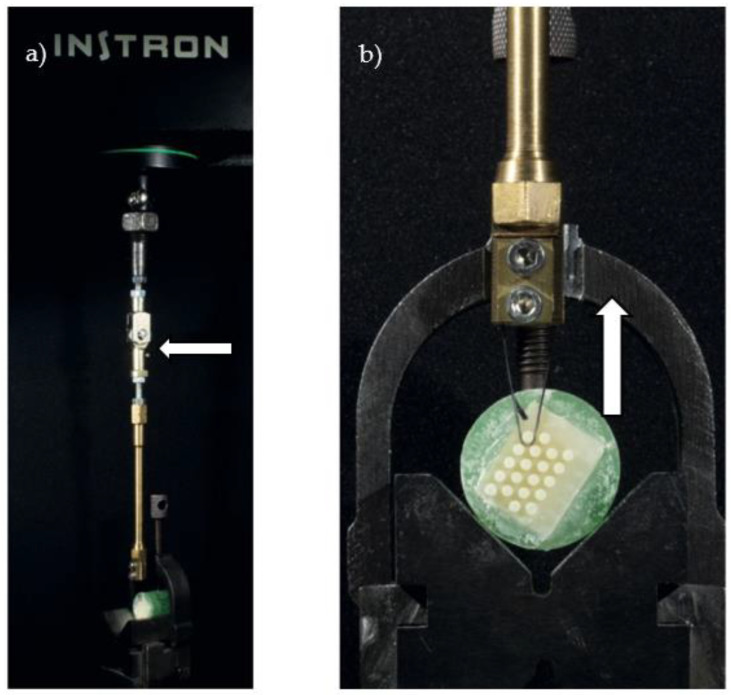
**Representation of the mSBS test.** (**a**) A pulling arm with a flexible ball-joint connection (white arrow) was fixed to the Instron machine load cell. (**b**) A zirconia ceramic tablet with an array of 20 CBC cylinders adhered to its surface was partially embedded in a green resin cylinder mounted on a V-shaped holder. A 0.30 mm thick steel wire is pulling the first CBC cylinder in an upward direction, generating shear stress on the adhesive interface in parallel to the ceramic plane.

**Figure 3 polymers-14-05301-f003:**
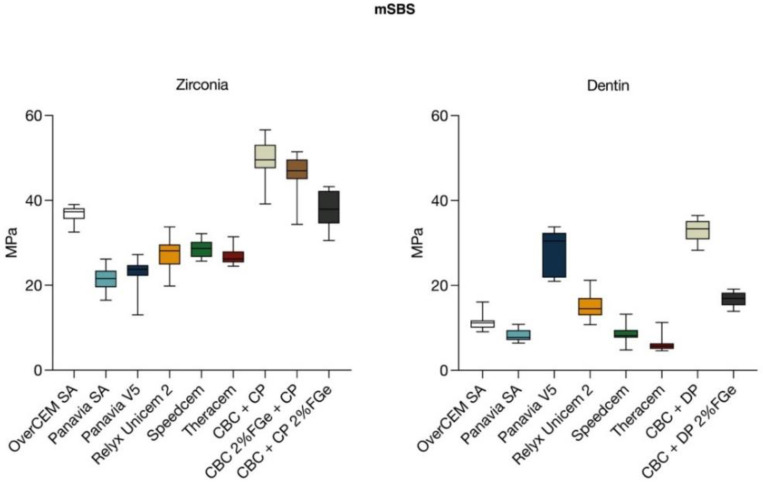
Graphical analysis of values reached by tested materials for mSBS to zirconia and dentin substrates.

**Figure 4 polymers-14-05301-f004:**
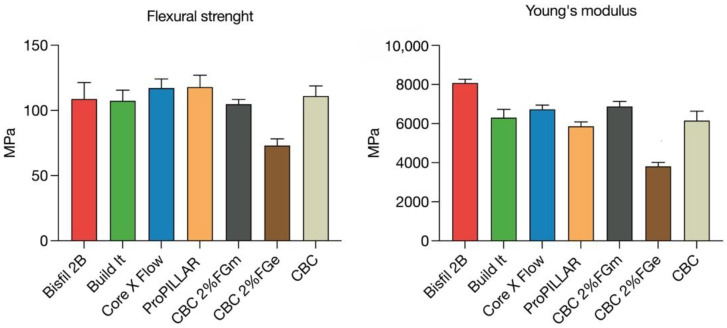
Graphical analysis of values reached by tested materials in flexural strength tests.

**Figure 5 polymers-14-05301-f005:**
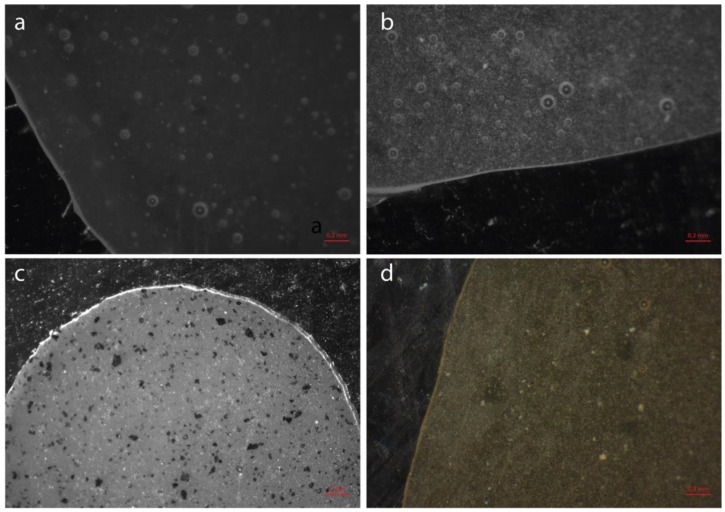
Light microscope images of (**a**) base paste and (**b**) catalyst of the CBC before degasification showing the composite fillers embedded in the resin matrix. (**c**) Image of base paste of CBC 2% FGm under optical microscope; note the relatively large size of the micronized fluorographene and graphene clusters (black particles). (**d**) Image of catalyst paste of CBC 2% FGe; exfoliated FG clusters are far beyond the optical microscope resolution.

**Figure 6 polymers-14-05301-f006:**
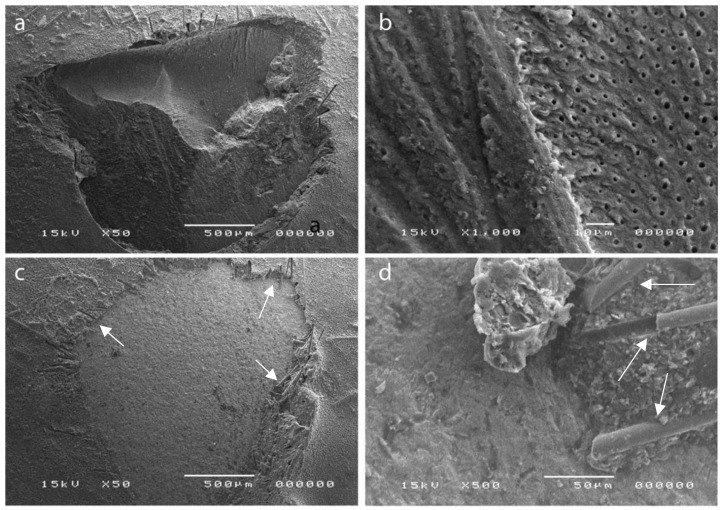
CBC prototype and detachment sites of the cylinders after the mSBS tests: (**a**,**b**) mixed adhesive and cohesive fractures on dentin; (**c**,**d**) adhesive fractures on zirconia with short glass fibers of about 14 µm in diameter clearly visible (white arrows).

**Table 1 polymers-14-05301-t001:** Compositions of the materials used in this study. Fluorographene was added to these materials after micronization (CBC 2% FGm) and exfoliation (CBC 2% FGm, DP 2% FG, and CP 2% FG). Abbreviations: UDMA: urethane dimethacrylate; TEGDMA: tetraethyleneglycol dimethacrylate; Bis-GMA: bisphenol A-glycidyl methacrylate; HEMA: hydroxy ethyl methacrylate; 10-MDP: 10-methacryloyloxydecyl dihydrogen phosphate; BPO: benzoyl peroxide; BHT: butylated hydroxytoluene; CQ: camphoroquinone; YbF_3_: ytterbium fluoride; C_3_H_6_O: acetone.

Material	Batch No.	Composition
CBC	170321B	UDMA, TEGDMA, Bis-GMA, HEMA, 10-MDP, BPO, BHT, CQ, YbF3, glass fillers
Dentin Primer	DP260321	10-MDP, H_2_O, C_3_H_6_O, HEMA
Ceramic Primer	CP260321	10-MDP, ethanol, silane (3-trimethoxysilylpropyl methacrylate)

**Table 2 polymers-14-05301-t002:** Materials used for mSBS test.

Material (*n* = 20)	Batch Number	Manufacturer	Dentin Surface Treatment	Zirconia Surface Treatment
OverCEM SA (Self-adhesive cement)	TRS1119	Overfibers s.r.l., Imola, Italy	400-grit SiC paper + clotramid (30 s) − water − air	Al_2_O_3_ sandblasting (50 μm, 3.5 bar pressure, 10 mm distance)
Panavia SA (Self-adhesive cement)	4A0010	Kuraray Europe GmbH, Hattersheim am Main, Germany	400-grit SiC paper + clotramid (30 s) − water − air	Al_2_O_3_ sandblasting (50 μm, 3.5 bar pressure, 10 mm distance)
Panavia V5 + Tooth Primer (Adhesive cement)	950057	Kuraray Europe GmbH, Hattersheim am Main, Germany	400-grit SiC paper + clotramid (30 s) − water − air + tooth primer (20 s) − wait (20 s) − air (20 s)	Al_2_O_3_ sandblasting (50 μm, 3.5 bar pressure, 10 mm distance) + ceramic primer (15 s) − wait (15 s) − air (15 s)
Relyx Unicem 2 (Self-adhesive cement)	6026795	3M ESPE Dental Products, Saint Paul, MN, USA	400-grit SiCpaper + clotramid (30 s) − water − air	Al_2_O_3_ sandblasting (50 μm, 3.5 bar pressure, 10 mm distance)
Speedcem (Self-adhesive cement)	Y10129	Ivoclar Vivadent AG, Schaan, Liechtenstein	400-grit SiC paper + clotramid (30 s) − water − air	Al_2_O_3_ sandblasting (50 μm, 3.5 bar pressure, 10 mm distance)
Theracem (Self-adhesive)	1900007482	Bisco, Inc., Schaumburg, IL, USA	400-grit SiCpaper + clotramid (30 s) − water − air	Al_2_O_3_ sandblasting (50 μm, 3.5 bar pressure, 10 mm distance)
CBC + primers (Control)	170321B + CP260321 + DP260321	-	400-grit SiCpaper + clotramid (30 s) − water − air + DP (15 s) − wait (15 s) − air (15 s)	Al_2_O_3_ sandblasting (50 μm, 3.5 bar pressure, 10 mm distance) + CP (15 s) − wait (15 s) − air (15 s)
CBC 2% FG + primer	071021 + CP260321 + DP260321	-	Not carried out due to early polymerization of the material	Al_2_O_3_ sandblasting (50 μm, 3.5 bar pressure, 10 mm distance) + CP (15 s) − wait (15 s) − air (15 s)
CBC + primers 2% FG	170321B + CP210921 + DP210921	-	400-grit SiC paper + clotramid (30 s) − water − air + DP 2% FGe (15 s) − wait (15 s) − air (15 s)	Al_2_O_3_ sandblasting (50 μm, 3.5 bar pressure, 10 mm distance) + CP 2% FGe (15 s) − wait (15 s) − air (15 s)

**Table 3 polymers-14-05301-t003:** Materials submitted to the flexural strength test.

Materials	Batch No.	Manufacturer	Sample Size
Bisfil 2B	4A0010	Bisco, Inc., Schaumburg, IL, USA	5
Build It	950057	Kerr Corporation, Orange, CA, USA	5
Core X Flow	Y10129	Ivoclar Vivadent AG, Principality of Schaan, Liechtenstein	5
ProPILLAR	1900007482	P.L. Superior Dental Materials Gmbh, Hamburg, Germany	5
CBC	170321B		5
CBC 2% FGm	060521		6
CBC 2% FGe	071021		4

**Table 4 polymers-14-05301-t004:** Results of mSBS and flexural strength tests.

Adhesive Properties			Mechanical Properties		
Material	mSBS on Zirconia (MPa)	mSBS on Dentin (MPa)	Material	Flexural Strength (MPa)	Young’s Modulus (MPa)
OverCEM SA	36.84 ± 1.7	11.48 ± 2.0	Bisfil 2B	108.9 ± 12.57	8081 ± 189.9
Panavia SA	21.38 ± 2.7	8.25 ± 1.3	Build It	107.4 ± 8.2	6307 ± 419.4
Panavia V5	23.13 ± 3.1	28.12 ± 5.0	Core X Flow	117.1 ± 7.0	6730 ± 218.9
Relyx Unicem 2	27.22 ± 3.7	15.12 ± 2.9	ProPILLAR	118.0 ± 9.1	5855 ± 227.1
Speedcem	28.66 ± 2.0	8.55 ± 2.0	CBC	111.1 ± 7.8	6155 ± 481.5
Theracem	26.95 ± 2.3	6.20 ± 1.9	CBC 2% FGm	104.8 ± 3.7	6876 ± 261.1
CBC + primers	49.73 ± 4.7	32.89 ± 2.4	e-CBC 2% FGe	104.8 ± 3.7	3814 ± 202.5
CBC 2% FGe + primers	46.53 ± 4.0	-			
CBC + primers 2% FGe	37.90 ± 4.0	16.74 ± 1.7			

## Data Availability

The data supporting the conclusions are included in the manuscript.

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
