# Peer review of "Effect of Fluorographene Addition on Mechanical and Adhesive Properties of a New Core Build-Up Composite"

_polymers, 2022, doi:10.3390/polym14235301_

Round 1

Reviewer 1 Report

How about if you put a post on it? If it's a prototype, I'd like to request that much.

Author Response

First of all, we would thank the Reviewers for the time spent in revising the manuscript. All the comments were specifically addressed and now we hope the manuscript would result improved to be taken into consideration for publication.

Answer to Reviewer #1

Reviewer: How about if you put a post on it? If it's a prototype, I'd like to request that much.

Our response: Thank you very much for the suggestion. It should be mentioned that further investigations are currently ongoing on this topic.

Reviewer 2 Report

This is a very premature manuscript. Clearly, there are many concerns with this manuscript:

1. A clear objective and innovation of this work needs to be addressed in the introduction section.

2. The author needs to rationalize the usage of fluoregraphene as additive. What is the purpose of SEM imaging on fluoregraphene? Does the multilayered structure affect the mechanical property of of CBC?

3. The legend of Figure 5 is not readable at all. 

4. There are two conclusion section.

5. Comments from the author is not erased yet.

There are many more concerns beyong above comments. This manuscript needs tremendous revisions before being considered for the journal. 

Author Response

First of all, we would thank the Reviewers for the time spent in revising the manuscript. All the comments were specifically addressed and now we hope the manuscript would result improved to be taken into consideration for publication.

Answer to Reviewer #2:

This is a very premature manuscript. Clearly, there are many concerns with this manuscript:

Reviewer: A clear objective and innovation of this work needs to be addressed in the introduction section.

Our response: We thank the Reviewer for pointing out this important information. Accordingly, the Introduction section was re-formulated focusing on highlight the main objective of the study.

Reviewer: The author needs to rationalize the usage of fluoregraphene as additive. What is the purpose of SEM imaging on fluoregraphene? Does the multilayered structure affect the mechanical property of of CBC?

Our response: Thank you to the Reviewer for the nice comment. Accordingly, we have added additional information regarding the addition of fluorografene into the CBC material and the dentin and ceramic primers both in the Introduction and Discussion sections. In particular, the objective of SEM imaging was intended to morphologically characterize industrial fluorografene to provide additional information on the structure, as no much data exists in the dental field. Therefore, the multilayer structure, as confirmed by SEM images are intended to comply with the purpose of our study, that is add FG in dental materials, to increase the mechanical properties and resistance of the materials themselves. As outlined in the paper, this is a preliminary paper and further studies are necessary to confirm these data.

Reviewer: The legend of Figure 5 is not readable at all. 

Our response: Legend of Fig. 5 has been modified and implemented with additional information, as requested.

Reviewer: There are two conclusion section.

Our response: Thank you to reviewer for the comment. We agree about the presence of the double conclusions and we re-arranged the paragraphs, accordingly.

Reviewer: Comments from the author is not erased yet.

Our response: We are afraid about this inconvenient. We have double-checked the manuscript and erased all comments and revisions.

Reviewer 3 Report

Many thanks for the difficult scientific work carried out. The article touches on a very relevant and important topic today. The scientific work made a pleasant impression on me. The article is easy to read despite the large experimental data. I am sure that the results will be of interest to a wide range of readers. 

However, I wanted to comment on a couple of points that I think should be considered.

1. Please add references for the "Introduction" part of text (lines 29-41).

2. Lines 43-45 meaningless text. The information about of host material (matrix) is fully absent. 

3. Line 44: The successfulness formation of chemical bonding in hydroxyapatite also depend on crystallinity degree, porosity and impurity ions. Please reconsider a little bit . 

4. Perhaps you should add more detailed information for construction on Figure 2, because without a description, this figure will not informative for the readers and can simply be deleted or added to the Supplemental section.

5. 2.4 Optical and SEM analysis section, line 191. Please add all experimental and set up conditions. All mentioned experiments in the article must be reproducible.

6. line 213, Is the p = 0.0001 real limit value to use its as statistical significance? Clinical studies sometimes with difficulty at p = 0.05 achieve statistical confirmation of their hypothesis.

7. There is not enough information about the physical and chemical type of processes to improve the degree of adhesion. For example, for calcium phosphate materials (hydroxyapatite or tricalcium phosphate), an improvement in the degree of adhesion occurs due to Al3+ or Mg2+ impurity centers, which leads to a defective surface structure and, as a consequence, an improvement in the "binding" of bone tissue.

8. The biological compatibility of the new improved materials is also important. Does the procedure of sample synthesis lead to an increase in toxicity?

9. Please design the article strictly according to the requirements of MDPI journals.

I will recommend this article for publication in the MDPI Polymers journal after reviewing the listed comments.

Author Response

First of all, we would thank the Reviewers for the time spent in revising the manuscript. All the comments were specifically addressed and now we hope the manuscript would result improved to be taken into consideration for publication.

Reviewer #3:

Reviewer: Please add references for the "Introduction" part of text (lines 29-41).

Our response: Thank you for the suggestion. Proper references have been added were necessary.

Reviewer: Lines 43-45 meaningless text. The information about of host material (matrix) is fully absent. 

Our response: Thank you for the comment. The text has been changed accordingly to the reviewer comments.

Reviewer: Line 44: The successfulness formation of chemical bonding in hydroxyapatite also depend on crystallinity degree, porosity and impurity ions. Please reconsider a little bit . 

Our response: The sentence related to 10-MDP has been removed and commented in the discussion section as suggested by the Reviewer.

Reviewer: Perhaps you should add more detailed information for construction on Figure 2, because without a description, this figure will not informative for the readers and can simply be deleted or added to the Supplemental section.

Our response. As suggested, more detailed information has been added to Fig.2.

Reviewer: Optical and SEM analysis section, line 191. Please add all experimental and set up conditions. All mentioned experiments in the article must be reproducible. 

Our response: We agree with the Reviewer regarding the need of reproduce laboratory experiments. Accordingly, we implemented the stereomicroscope and SEM section with further information regarding sample preparation.

Reviewer: Line 213, Is the p = 0.0001 real limit value to use its as statistical significance? Clinical studies sometimes with difficulty at p = 0.05 achieve statistical confirmation of their hypothesis.

Our response: We thank the Reviewer for the interesting comment. In the present work, the statistical differences between the tested groups were rather pronounced, with the p values lower than 0.0001, reaffirming a high extent of differences between the tested groups. We applied the p-value threshold used and recommended in a vast number of research fields. There has been a lot of debate recently whether the 0.05 value of p introduced by Pearson in 1900 is enough to claim statistical significance, and whether this threshold should be elevated. The majority the research community though supports still the use of the threshold of 0.05, since the more stringent requirements in terms of p-value could require sample sizes that are unattainable and would discourage a large number of researchers from engaging into research due to lack of funds and organizational possibilities. This problem was raised even at the possibility of changing the p-value threshold to 0.05. Increasing it to 0.0001 would indeed be devastating for the research community. However, we believe that engaging deeper into the discussion on whether the p-value threshold should be elevated or not, is out of the scope of this manuscript and beyond the limits of statistical knowledge of the authors.

Reviewer: There is not enough information about the physical and chemical type of processes to improve the degree of adhesion. For example, for calcium phosphate materials (hydroxyapatite or tricalcium phosphate), an improvement in the degree of adhesion occurs due to Al3+ or Mg2+ impurity centers, which leads to a defective surface structure and, as a consequence, an improvement in the "binding" of bone tissue.

Our response: Following Reviewer’s comment, we reformatted the discussion section accordingly in order to implement information on this issue.

Reviewer: The biological compatibility of the new improved materials is also important. Does the procedure of sample synthesis lead to an increase in toxicity?

Our response: The main goal was to develop a CBC with dedicated adhesives and to assess the  effects of 2% fg addition on the adhesive and flexural properties of the cbc/adhesives prototype, not to investigate on its toxicity. However, the topic is of great interest: actually, toxicity of fluorographene has been evaluated in some studies, that have been included in the revised manuscript along with a short discussion and data interpretation. It appears that toxicity to human derived cells is dose-dependent, and also influenced by the fluorination degree. Considering the extremely low dissolution rate of resin dental cements and the tiny FG wt% in the CBC and adhesive primer masses (less than 0.7%), on the basis of the reported studies it seems unlikely that FG could induce serious adverse biological effects. Certainly, further studies should be performed to deeply evalute the toxicity of FG when added to dental composite chemistry and its interactions with the other components of the new CBC and primer system.

Reviewer: Please design the article strictly according to the requirements of MDPI journals.

Our response: The guidelines of the Journal have been revised and the paper modified accordingly.

Round 2

Reviewer 2 Report

All my concerns are solved now and I think the current manuscript is quanlified for pulication. 

Reviewer 3 Report

I recommend this material for further processing by editors and publication in the journal.